# Exosome Composition and Seminal Plasma Proteome: A Promising Source of Biomarkers of Male Infertility

**DOI:** 10.3390/ijms21197022

**Published:** 2020-09-24

**Authors:** Luz Candenas, Rosanna Chianese

**Affiliations:** 1Instituto de Investigaciones Químicas, CSIC, Avenida Américo Vespucio 49, 41092 Sevilla, Spain; luzcandenas@iiq.csic.es; 2Department of Experimental Medicine, University of Campania Luigi Vanvitelli, via Costantinopoli 16, 80138 Napoli, Italy

**Keywords:** sperm, seminal plasma, seminal exosomes, infertility, seminal proteins, biomarkers

## Abstract

Infertility has become a global health issue, with approximately 50% of infertility cases generated by disorders in male reproduction. Spermatozoa are conveyed towards female genital tracts in a safe surrounding provided by the seminal plasma. Interestingly, this dynamically changing medium is a rich source of proteins, essential not only for sperm transport, but also for its protection and maturation. Most of the seminal proteins are acquired by spermatozoa in transit through exosomes (epididymosomes and prostasomes). The high number of seminal proteins, the increasing knowledge of their origins and biological functions and their differential expression in the case of azoospermia, asthenozoospermia, oligozoospermia and teratozoospermia or other conditions of male infertility have allowed the identification of a wide variety of biomarker candidates and their involvement in biological pathways, thus to strongly suggest that the proteomic landscape of seminal plasma may be a potential indicator of sperm dysfunction. This review summarizes the current knowledge in seminal plasma proteomics and its potentiality as a diagnostic tool in different degrees of male infertility.

## 1. Introduction

Spermatozoa (SPZ) are streamlined cells, optimized to propagate their haploid genome by exploiting the egg. They are equipped with a tail in order to acquire speed and efficiency in the task of fertilization. To this end, during their maturation along the epididymis, SPZ lose most cytoplasmic organelles, which are unnecessary for their mission. SPZ travel through the female genital tract helped by fluids, which are produced by different male sex organs, forming the seminal plasma (SP) [1]. Normal semen is a mixture of approximately 5% SPZ (and other accompanying cells, collectively referred to as round cells) and 95% SP [2,3].

SP, the non-cellular liquid component of semen, is formed by secretions from the testis, the epididymis, the prostate, the Cowper (bulbourethral) and Littre’s (periurethral) glands and the ampulla of the ductus deferens, with a major contribution (about 65% of the total semen volume) of the seminal vesicles [2,3,4,5,6,7,8,9,10].

Sugars, oligosaccharides, glycans, lipids, inorganic ions (calcium, magnesium, potassium, sodium and zinc) and small metabolites are abundant components of the SP. Galactose and sialic acid are mainly secreted by the bulbourethral glands; the prostate is the main source of citric acid, inositol, calcium, zinc and magnesium found in the ejaculate and the ampulla of the vas deferens is the principal source of sorbitol in men [5,7]. Secretions from the seminal vesiclesare rich in ascorbic acid, prostaglandins and fructose, a major sperm nutrient, while the concentrations of L-carnitine and glycerylphosphorylcholine are indications of epididymal function [3,5,7,8,9,10].

What is interesting is that SP is particularly enriched in proteins, RNAs and lipids that are present in the fluid and/or encapsulated in extracellular vesicles or exosomes. All these components are mixed in a heterogeneous manner, with a composition that varies, not only among and within individuals, but also even within an ejaculate [2,9,11]. Since the complexity of the SP composition, it is evident that it has not just the function of sperm transport [2]. In fact, along both male and female reproductive tracts, SPZ encounter a novel extracellular milieu, able to modulate their metabolism, their intracellular structures and biochemical composition, through an intense communication that mainly occurs via exosomes [12].

Exosomes are membranous nanovesicles of 30–150 nm in diameter, for many years considered as organelles to remove cell debris [13,14], difficult to isolate as pure populations. In 1996, Raposo et al. suggested that they were involved in immunological processes [13]. Currently, it is accepted that exosomes play key roles in intercellular communications, reaching target cells through body fluids [14,15,16].

Along the reproductive tracts, a major breakthrough in exosome research has been the deep characterization of exosomal cargo—by means of the advancement of RNA sequencing and high resolution mass spectrometry methodologies—that is rich in messenger RNAs (mRNAs), non-coding RNAs (ncRNAs) and proteins [1]. Considering that SPZ are transcriptionally quiescent cells, the acquisition of new secretory products during the post-testicular maturation along the epididymis—ascribed to exosomes of epididymal origin, known as epididymosomes—confers to SPZ a new molecular guise [12,17,18,19,20]. However, sperm travel still continues in female reproductive tracts, where the exosomes play an important role through the activation of inflammatory and immune mechanisms [21].

Although the transfer of proteins to SPZ represents a common feature of seminal exosomes, it is actually discussed whether distinct populations of vesicles carry out different tasks [22,23]. Whatever the case, the analysis of seminal proteome has corroborated the involvement of this fluid in the control of sperm physiology [24], motility [25], morphology [26], concentration [27] and protection from oxidation [28]. Interestingly, Ca^2+^ signaling machinery required for sperm motility may be directly achieved from SP [29]. Similarly, seminal exosomes may regulate the tyrosine phosphorylation pattern, a key intracellular signaling event controlling sperm capacitation [30].

The regulation of all these sperm functions surely depends on the ability of exosomes to bind to SPZ membrane and fuse with them; several conditions—such as pH, temperature and zinc concentrations—have been shown to influence the exosomal uptake by SPZ [31,32,33]. However, SPZ still remain receptive to exosome-shuttled cargos after ejaculation, benefiting from the exosomal transfer of key proteins, in terms of motility, capacitation and quality [34].

Remarkably, the seminal proteomic profile may be indicative of genital tract dysfunctions and, therefore, serve as a biomarker of infertility [3,10,35,36]. In this regard, although exosomes isolated from SP of normo-, astheno- and azoospermic patients display similar features in terms of shape, size and expression of typical exosomal markers, they differ in their cargo. In fact, just normozoospermic-derived exosomes are able to improve SPZ motility and trigger capacitation [34].

Several questions still remain unresolved. Much effort is required to deeply unveil (i) the molecular composition of SP and, particularly, of seminal exosomes, (ii) who regulates vesicle trafficking and fusion with SPZ and (iii) which are the precise exosomal functions in sperm physiology.

## 2. Exosomes in the Male Reproductive Tract: Their Potential Role in Sperm Physiology

Outside the testis, SPZ travel along three segments of the epididymis (in Greek, epi for ‘on’ and didumoi for ‘testis’): the proximal caput, the elongated corpus and the distal cauda, characterized by a regionalized pattern of gene and protein expression [37,38], with the caput epididymis being the most active in protein synthesis and secretion [39].

The analysis of luminal fluid composition has revealed a complex macromolecular landscape encompassing soluble factors and exosome-like vesicles. A fine crosstalk takes place between SPZ in transit and the epididymal epithelium. How such a crosstalk happens has been a question that has intrigued reproductive biologists since the 1970s. Apocrine secretion has been suggested as the involved mechanism; it implicates the formation of cytoplasmic blebs at the apical pole of the principal epithelial secretory cells that break down to release their contents, including membrane vesicles: the epididymosomes [40,41]. These vesicles were firstly described at the electron microscopic level by Yanagimachi et al., as involved in cholesterol transfer to sperm plasma membrane [42]. However, epithelial principal cells also regulate the composition of the intraluminal milieu by merocrine secretion [41].

Since mature SPZ are devoid of the lysosomal organelles—typical targets for endocytosed cargo [43]—and are unable to recycle lipids [44], the endocytic uptake is severely compromised and a putative mechanism for epididymosome delivery is the formation of transient fusion pores between epididymosomes and sperm, rather than a complete fusion of membranes [39]. Consequently, epididymosomal uptake may involve Rab and soluble N-ethylmaleimide-Sensitive Factor (NSF) attachment protein receptor (SNARE) proteins, identified in both SPZ and epididymosomes by proteomic analysis [45,46]. Other signaling factors may also mediate such a process; for example, dynamin has been suggested to control the rate of fusion pore expansion [39].

Interestingly, each epididymal segment produces a heterogeneous population of epididymosomes, with different lipid and protein composition [45,46,47]. From caput to cauda epididymis, epididymosomes change their lipid content, with an increase in sphingomyelin and a decrease in cholesterol [48]. The proteins associated with epididymosomes characterize two distinct vesicle populations: the first one—enriched in CD9 and other tetraspanin proteins—interacts with live SPZ; the second one—characterized by the sperm binding protein 1 (ELSPBP1)—interacts with dying/dead SPZ [49]. Defective SPZ can also be targeted by ubiquitin (UBQ), an epididymosome-associated protein that SPZ receive during their transit [50]. Remarkably, epididymosome–sperm interaction is highly selective: SPZ in transit acquire from the intraluminal compartment of epididymis proteins deprived of the N-terminal signal peptide and associated with glycosylphosphatidylinositol (GPI)-anchor or biotinylated [46,51]. Thus, the GPI-anchor-mediated docking is a way of epididymosomal delivery cargo [39]. SPZ preferentially sequester proteins into the post-acrosomal sheath, at the level of specific membrane subdomains: the rafts, lipid domains involved in signaling pathways and protein/lipid trafficking [52]. As outlined in Figure 1, proteins are transferred from rafts of epididymosomes to rafts of maturating SPZ [53], in a temperature- and pH-dependent manner and under zinc stimulation [54]. Ionic composition of epididymal fluid—characterized by a lower overall concentration of Na^+^, Cl^−^ and HCO_3_^−^—strongly regulates luminal acidification that helps keep SPZ in a dormant state [55]. Epithelial clear cells lead to further acidification of the epididymal luminal environment by pumping protons via ATPase-dependent pumps [55].

Starting from the 1960s important sperm functions, such as the acquisition of sperm progressive motility and fertilizing capacity, were hypothesized under the control of the epididymis [56,57]. Considering that it is not surprising that epididymosomes convey to SPZ a large cargo of proteins encompassing key classes of enzymes, chaperones and structural proteins; for example, the cysteine-rich secretory protein 1 (CRISP1) regulates Ca^2+^ channels in the sperm membrane [58], the sperm adhesion molecule 1 (SPAM1, also known as PH-20) is involved in sperm-zona pellucida adhesion [59] and the macrophage migration inhibitory factor (MIF) actively participates in sperm motility acquisition [60]. Epididymosomes also contain proteins providing antimicrobial and proteolytic attack protection [12], and glutathione peroxidase (GPX) enzymes preventing premature acrosome reaction and protecting sperm from oxidative stress [61,62].

Along the epididymis, sperm proteins undergo several post-translational modifications, despite SPZ being silent cells [57]. This is the case of two proteins, both required for gamete interaction: IZUMO1—whose cytoplasmic domain becomes heavily phosphorylated in the epididymis—and ADAM3 (also known as cyritestin). The sperm membrane also acquires greater fluidity, by incorporating unsaturated fatty acids and gradually removing cholesterol and phospholipids [7]. Despite these changes occurring in the epididymal environment, sperm is still in a quiescent state by means of the acidic pH, as described above, and the “decapacitation” factors, such as the binder of sperm protein (BSP), that binds to the sperm membrane in order to protect SPZ until they reach the fallopian tube to achieve capacitation [63,64].

During ejaculation, SPZ encounter another subset of exosomes: the prostasomes, firstly reported in the 1970s [65], as vesicles with a size from 30 to 500 nm, secreted from the acinar lumen of prostate epithelial cells, together with soluble components. Prostasomes are contained within multivesicular bodies and their secretion into the extracellular prostatic fluid requires the fusion of membranes.

Prostasomal lipid composition especially includes saturated or monounsaturated fatty acids, with an unusually high cholesterol content and sphingomyelin, probably to account for the very high stability and rigidity of the prostasomal membrane [66]. This unusual lipid composition is a major difference between prostasomes and sperm cells, thus to suggest that prostasomes may transfer cholesterol and other saturated lipids to sperm after fusion, leading to a decrease in the fluidity of the sperm membrane that prevents an early capacitation and a premature acrosome reaction [62,66].

Prostasomes are the most abundant exosomes in SP and, depending on their size and molecular composition, two different populations can be distinguished: smaller and larger vesicles, enriched in glioma pathogenesis-related 2 (GLIPR2) and annexin A1 (ANXA1) proteins, respectively [67]. These vesicles also contain enzymes, signal transduction proteins, chaperone proteins, transport and structural proteins, GTP-binding and other signaling proteins [68,69], high concentrations of prostatic-specific acid phosphatase (ACP3 or ACPP, usually known as PAP), prostate-specific antigen (KLK3, usually known as PSA), type 2 transmembrane serine protease (TMPRSS2), prostate-specific transglutaminase (TGM4) and prostate stem cell antigen (PSCA) [69,70]. Prostasomes are also able to produce adenosine triphosphate (ATP), which suggests an energy metabolic function associated with these extracellular vesicles [71].

Prostasomes are rich in divalent cations such as Ca^2+^, Zn^2+^ and Mg^2+^ that—transferred from prostasomes to SPZ through a pH-dependent fusion mechanism—contribute to modulate sperm ion concentrations and flagellar motility [72,73]. As outlined in Figure 1, the recruitment of prostasomes by SPZ requires the presence of bicarbonate and an environment with pH 7.5 [33]; live sperm cells recruit prostasomes—with a high binding specificity—primarily at their head region [5]. The best conditions for prostasome recruitment should occur in the uterus since the neutral pH and the local presence of bicarbonate represent favorable conditions for prostasome binding, but not fusion [33]. Therefore, it is plausible that SPZ may enter the oviduct with prostasomes draped over their cell surface; then, in proximity to the oocyte–cumulus complex, prostasomes may fuse with SPZ, favored by acidic conditions [74], integrating in them cytosolic and membrane constituents.

Prostasomal proteins have antioxidant, antimicrobial and antibacterial properties; coagulant properties serve to prevent sperm contact with female blood, and protect SPZ against the immune response in the female reproductive tract by inhibiting monocyte and neutrophil phagocytosis [22,64,75]. Additionally, these vesicles play key roles, both inhibitory and stimulatory, in several physiological sperm functions, such as sperm motility, sperm capacitation and acrosome reaction [62,76]. Along the female reproductive tract, SPZ are mature, but still unable to fertilize the oocyte. They undergo capacitation, acquire hyperactive motility and change their membrane composition through the stripping and/or modification of many glycoproteins that coat the sperm surface and the bidirectional movement of phospholipids across the lipid bilayer [7,77]. This change is stimulated by bicarbonate that rapidly activates sperm-specific adenylate cyclase (sACY), leading to the generation of cAMP and an increase in phosphorylation of proteins in serine, threonine and tyrosine residues [78,79,80,81]. Prostasomes help to increase intracellular cAMP levels [82]. During sperm cell capacitation, there is an activation of epithelial and voltage Na+ channels and of K+ channels, with subsequent hyperpolarization of the plasma membrane, increase in pH and opening of Ca^2+^ channels [79,80,81,83].

Interestingly, Ca^2+^ signaling—sustained by prostasomes [5]—influences both sperm motility and acrosome reaction that takes place under the stimulation of progesterone secreted by the cumulus cells surrounding the oocyte [84]. Ca^2+^ signaling factors (including receptors, enzymes and membrane Ca^2+^ channels, mainly annexins) and several enzymes making SPZ more sensitive to progesterone effects are provided by prostasomes and are required to mediate the effect of progesterone on the acrosome reaction [29]. In this regard, under physiological conditions, increased progesterone levels stimulate prostasome and sperm membrane fusion [5]. Furthermore, prostasomes are able to prevent an early acrosomal reaction by transferring to sperm membrane cholesterol and sphingomyelin [22].

## 3. Proteomic Landscape of SP: A Potential Indicator of Sperm Dysfunction

Infertility is a condition defined as the inability of a couple to conceive after at least one year of unprotected intercourse. This pathology has become a global health issue with a general prevalence of 15%, affecting one out of six couples of reproductive age. According to global statistics, male infertility is the cause of approximately 50% of infertility cases, either as a sole cause or in combination with a female infertility factor [85]. Semen analysis is the most widely technique used to determine male fertility, but it has become evident that, in many cases, a basic seminogram is insufficient to determine the fertility status of the male partner. In this context, the SP and, particularly, its proteome profile, offers a great opportunity to increase our knowledge of the physiology of the male reproductive system and for the discovery of biomarkers to improve diagnosis or classification of a wide range of male fertility disorders [3,22,23,35].

SP was for a long time considered a passive medium just needed for delivering sperm to the female reproductive tract. However, the recent advances in proteomic technologies have permitted the identification of thousands of proteins and, as described above, there is now little doubt that SP has important effects over sperm function and male fertility [2,3,35,86,87,88,89]. SP proteins participate in the formation of the seminal coagulum and in the liquefaction process due to its content on semenogelins (SEMG), fibronectin (FN1) and KLK3. Semenogelin-1 (SEMG1) and Semenogelin-2 (SEMG2) are the most abundant proteins in SP and are the main components of the coagulum formed at ejaculation, which inhibits premature sperm capacitation and, due to its buffering properties (pH 7.35–7.50), provides a protective environment for ejaculated SPZ against the acidic milieu of the vagina [90].

SP proteins are also involved in the regulation of sperm motility, hyperactivation and acrosomal reaction; they provide protection against oxidative processes and have immunosuppressive properties inducing an effective inhibition of lymphocytes, macrophages and the complement system [2,5,6,10,22,64,91,92,93]. SP has the additional advantage that it is usually obtained in large amounts, providing a good source for proteomic analysis. Furthermore, the presence of tissue-specific molecular mediators should permit an identification of the origin of male infertility [3,10,11,22,27,35].

### 3.1. The SP Proteome

SP is predominantly made up of proteins with a range of concentrations between 35 and 55 mg/mL [3,9,10]. The bulk of the proteins found in semen derive from the seminal vesicles, with SEMG1, SEMG2, FN1 and lactotransferrin (LTF) being the predominant proteins secreted by these glands [4,86]. Nitric oxide synthase (NOS1), protease inhibitors such as the protein C-inhibitor (PCI) serpin and diverse cytokines, including transforming growth factor-β (TGFB) are also secreted by seminal vesicles [86,94]. However, one of the most abundant proteins in SP, albumin (ALB), that represents about one third of the protein content of semen, is, in part, of prostatic origin [95,96]. The prostate is also the main source of glutamate carboxypeptidase 2 (also known as folate hydrolase, FOLH1, or prostate-specific membrane antigen, PSMA), arachidonate 15-lipoxygenase B (ALOX15B), glutamine γ-glutamyltransferase 4 (TGM4), prostate-specific protein 94 (PSP-94, β-inhibin-β-microseminoprotein), ACPP and kallikrein hK3e (KLK3), the serine protease known as PSA [96,97]. Many other proteins identified in SP are of epididymal origin, including prostaglandin-H2 D-isomerase (also known as prostaglandin-D2-synthase or lipocalin-type prostaglandin-D-synthase; PTGDS), clusterin (CLU), epididymal secretory protein E1 (NPC2), MIF, human leukocyte antigen G (HLA-G), sperm protein P34H and a range of antioxidant-system proteins such as epididymal secretory glutathione peroxidase 5 (GPX5) and γ-glutamyltranspeptidase (GGT1) [3,41,86,98,99]. Transketolase-like protein 1 (TKTL1), phosphoglycerate kinase 2 (PGK2), L-lactate dehydrogenase C chain (LDHC), testis-expressed protein 101 (TEX101) and acrosomal protein SP1 (ACRV1) derive from the testicular secretion [10,100] while mucin (MUC), a protein extremely abundant in seminal fluid, is a protein characteristic of Cowper’s gland. Many proteins present in SP derive from various glands, such as ALB, which, in addition to the prostate, is originated in the testes and the epididymis; LTF, also originated in the epididymis, or prolactin inhibitory protein (PIP), whose origin is not only the seminal vesicle but also the testis or the epididymis [35,95]. KLK3, mainly released by the prostate, is also produced by the Littre’s gland [2] and SPAM1 (PH-20) protein is expressed in the epididymis and the testis [101]. Thus, the identified proteins come from the different organs involved in the formation of SP.

More recently, the tissue of origin of some seminal proteins has been determined by searching for tissue specificity in public databases [6,35,102]. However, in the particular case of the lowest expressed proteins, little is still known about their identity, origin and their precise physiological role. It must be considered that the 10 more abundant proteins in seminal fluid, including SEMG1, SEMG2 and ALB, represent approximately the 80% of all protein bulk mass, hindering the detection of the low abundance proteins [10,88].

### 3.2. Analysis of Human SP Proteome

The number of proteins detected in SP has been growing progressively with the advancement of available analytical techniques [3,10,11,35,103]. The initial studies of SP proteome identified the presence of a few proteins also abundant in blood serum, including ALB, α-, β- and γ-globulins as well as various phosphatases, aminopeptidases, glycosidases and hyaluronidase [104]. The use of high-resolution two-dimensional polyacrylamide gel electrophoresis (2D-PAGE) revealed the presence of a high number of protein spots and the existence of differences in some of these proteins between fertile and infertile men, allowing the immunological identification of some proteins abundant in SP, such as LTF, α-1-antitrypsin (SERPINA1), ACPP and creatine kinases (CK) [103,105,106]. More recently, the development of mass spectrometry (MS) techniques, including 2D-PAGE coupled with matrix-assisted laser desorption-ionization time-of-flight (MALDI-TOF) [107], combination of 1D-PAGE and liquid chromatography Fourier transform MS (LC–FTMS) [86] or liquid chromatography tandem mass spectrometry (LC–MS/MS) [87,88,100,102,108], has permitted the identification of more than 2000 proteins, revealing the great complexity of SP, which is additionally increased by the observation that many of these proteins exist in multiple post-transcriptional variants [3,6,91,103].

The study of common SP proteome in fertile men and the analysis of the origin and function of the found proteins could provide better insight into the sperm physiology, representing an essential step to identify novel biomarkers of male infertility. Recent large-scale proteomic analysis of human SP most frequently uses gene ontology and bioinformatic analysis to assign the cellular component, molecular functions and biological processes of the large number of proteins identified [86,87,88,89]. Pilch and Mann [86] performed the first large-scale proteomic analysis of human SP, identifying 923 proteins that were mainly involved in binding function, metabolism and the immune response. With respect to the molecular function, the authors observed that approximately 60% of the seminal fluid proteome was implicated in enzymatic activity. This included hydrolases and peptidases, which represent a high percentage (over 8%) of all identified proteins, as confirmed in other studies [100,102]. SP is also rich in protease inhibitors (over 4% of the identified proteins), with a predominance of the serine-type endopeptidase inhibitors known as serpins. The high number of proteases and protease inhibitors in SP evidences the importance of this system in this body fluid [86]. Another important functional group is formed by signal transduction molecules, representing more than 9% of all proteins, including Ras-related small GTPases, Rab and Rab-related proteins, which have been identified in prostasomes [68]. The seminal proteome also contains protein kinases, phosphatases, transporter proteins and structural molecules [86]. The precise function of many of these proteins remains to be determined.

In addition to soluble molecules and extracellular proteins, a substantial number of proteins identified in SP are cytoplasmic, associated with the great number of seminal exosomes present in this fluid. These vesicles can be collected from ejaculates by centrifugation, to remove the SPZ, followed by different ultracentrifugation steps of the supernatant [6,23,34,68,86]. This permits an analysis of seminal proteome associated exclusively with these extracellular vesicles. Utleg et al. [68] analyzed exosomes from five healthy men by using microcapillary HPLC–tandem mass spectrometry (µLC–MS/MS) and reported the presence of 139 proteins. The most abundant groups of proteins were: (a) enzymes (with special abundance of proteases such as ACPP, GGT1 and various members of the ec.3.4 subfamily, including aminopeptidase N (APN), aminopeptidase P (APP), neprilysin (MME), dipeptidyl peptidase IV (DPP4), FOLH1 and KLK3. APN and MME have been proposed to regulate human sperm cell motility through modulation of enkephalin and tachykinin levels [109,110]. (b) Transport/structural proteins, with an abundance of six members of the annexin family (ANXA1, 3, 5, 6 and 11) involved in Ca^2+^ regulation, membrane trafficking events, lipid reorganization in the membrane and endocytosis; α-actin (ACTA) and β-actin (ACTB), profilin I (PRO1) and II (PRO2), LTF, FN1 and tubulins α (TBA1) and β (TBB2). (c) GTP proteins, including proteins of the RAS family (RAB and RAP), involved in the regulation of vesicle trafficking. (d) Chaperone proteins including the heat shock-related proteins HSP27 (HS27), HSP70 (HS71, HS72 and HS76) and HSP90 (HS9A and HS9B) proteins; and (e) signal-transduction proteins including calmodulin (CALM), CLU, UBQ, zinc-alpha-2-glycoprotein (AZGP1, also known as ZA2G) and MIF.

A large-scale proteomic analysis has been recently performed [88] examining exosomes isolated from 12 healthy donors, randomly divided into two different pools, using high-resolution HPLC coupled with LC–MS/MS. A total of 1474 proteins were identified in all seminal exosomes samples. Enrichment analysis of GO terms revealed that, in term of cellular components, exosome-associated proteins were mostly linked to exosomes and cytoplasm and the main biological processes included metabolism, energy pathways, protein metabolism, cell growth and maintenance and transport, which is consistent with previous studies [34,69,86,87,102]. The proteins identified included exosome-marker proteins, such as HSP70, CD81, apoptosis-linked gene 2-interacting protein X (ALIX), ACTB, glyceraldehyde-3-phosphate dehydrogenase (GAPDH), phosphoglycerate dehydrogenase (PHGDH) and galectin-3-binding protein (LGALS3BP). It is remarkable that, despite an equivalent amount of total proteins from each individual in the two pools of healthy donors analyzed, there were large variations in the identified proteomes, suggesting the existence of great variations in seminal proteomes between individuals [67,88].

### 3.3. SP Candidate Biomarkers of Male Infertility

The high number of proteins, the increasing knowledge of their origin and biological function and the observation that some of them are differentially expressed between fertile and infertile men [3,10,11,22,23,27,35] have allowed the identification of a wide variety of SP proteins proposed as biomarker candidates for male infertility disorders. Some of these interesting candidate biomarkers and the infertility disorders to which they have been associated are shown in Table 1 and described in the following sections.

#### 3.3.1. SP Candidate Biomarkers for Azoospermia

Many of the actually proposed seminal biomarkers appear to be particularly useful to differentiate the causes of azoospermia.

In this disorder, the absence of SPZ in the semen may occur as a consequence of spermatogenesis abnormalities or inadequate hormonal stimulation, being known as non-obstructive (NOA), or be caused by seminal tract obstruction, the latter being known as obstructive azoospermia (OA) [10]. The finding of biomarkers to differentiate NOA and OA is very important to clarify the indication of testicular sperm extraction (TESE) and avoid repetitive multiple testicular biopsies as >50% of true NOA permit the recovery of testicular SPZ [10].

Initial observations showed that men with defects in seminal vesicles had a lower ejaculate volume, extremely low fructose levels and absence or traces of SEMG and LTF while presenting a high concentration of zinc and PSA in SP. Subsequent studies proposed blood serum inhibin-B as a potential biomarker of NOA [130]. Inhibin-B provides an accurate measure of Sertoli cell function and is a useful marker of spermatogenesis [7]; serum levels in OA were found to be similar to those in fertile donors while being significantly lower in NOA [128]. Within NOA patients, inhibin-B levels were significantly higher in patients with TESE+ (successful sperm recovery), in comparison with TESE- (failed sperm recovery) ones [130]. Subsequent studies proposed that the glycoprotein Anti-Müllerian hormone (AMH), produced by testicular Sertoli cells, could be a biomarker of azoospermia, as it was undetectable in SP of OA men and was present at very low levels in men with NOA, compared to fertile donors [131]. At present, both inhibin-B and AMH appear to have little specificity and sensitivity to be used as biomarkers [10,132]. Nevertheless, recent data suggest that they may be useful to predict sperm recovery after cryopreservation in men with asthenozoospermia [133].

Diamandis et al. [127] found that the concentration of the prostaglandin synthase PTGDS decreased progressively from normal to oligospermic to azoospermic patients and was almost absent in vasectomy patients. Median PTGDS levels in normal men were above 100-fold higher compared to vasectomized men. Next studies by this group confirmed that PTGDS could be used as a biomarker of OA, as its levels were significantly lower in these patients, compared to men with NOA [108,128].

By using 2D-PAGE MALDI-TOF and peptide sequence by tandem MS/MS, Starita-Geribaldi et al. [116] identified several proteins that were undetectable or present at low levels in the SP of azoospermic patients: CRISP1, CLU, NPC2, superoxide-dismutase (SOD) and serum amyloid p-component (SAP), the last one being undetectable in azoospermic patients with Sertoli cell-only syndrome. CRISP1 is an epididymal protein that binds to the postacrosomal region of the sperm head and plays a role in the regulation of sperm motility, acrosome reaction and sperm–egg fusion; its validity as a biomarker of azoospermia has been further confirmed by Western blot analysis [118] showing that CRISP1 is absent or present at very low levels in samples from patients with OA.

Yamakawa et al. [122] analyzed the seminal proteome of 10 fertile and 10 infertile men with azoospermia (7 NOA and 3 OA) using 2D gel electrophoresis and LC–MS/MS and identified four possible biomarkers for NOA: stabilin-2 (STAB2), 135 kd centrosomal protein (CP135), guanine-nucleotide-releasing protein (GNRP) and PIP, which were absent in samples from more than three patients with NOA, and one candidate marker for obstructive azoospermia, the epididymal protein NPC2 (Niemann-Pick disease C2 protein), absent in OA patients, with a possible application as a clinical marker to differentiate NOA from OA.

Batruch et al. [102] used strong-cation exchange LC–MS/MS to analyze the seminal proteome of five men with normal spermatogenesis, five men with NOA and five men who had undergone a vasectomy (PV), and identified more than 2000 proteins in SP. Among them, 32 proteins were only present in control individuals, 49 at lower abundance in PV, 3 unique to PV and 25 at higher abundance in PV. Since ejaculates from men who have undergone vasectomy are devoid of testicular and epididymal secretions, their seminal proteomes will derive from the prostate, seminal vesicles and periurethral glands. The proteins unique to control or at lower abundance in PV would thus have their origin in the testis and the epididymis. This remarks the importance of determining the tissue of origin of individual proteins in SP, as it would permit to identify the provenance of functional abnormalities in infertile subjects. In subsequent studies, this group identified differentially expressed proteins, such as gammaglutamyltransferase 7 (GGT7) and sorbitol dehydrogenase (SORD), found uniquely in NOA; PGK2, LDCH, histone H2B type 1-A (HIST1H2BA) and dipeptidase 3 (DPEP3), which were uniquely expressed in control samples and may thus be potential biomarkers of NOA and/or OA [102,108]. A panel of 18 biomarkers for differential diagnosis of NOA and OA has also been proposed, including the epididymis-expressed extracellular matrix protein 1 (ECM1) and the testis-expressed sequence 101 (TEX101), which showed a higher expression in NOA [102,124]. Additionally, the combined study of these two biomarkers, ECM1 and TEX101 permit the differentiation of different NOA subtypes, due to the specific expression of TEX101 in testicular germ cells, thus providing a non-invasive clinical assay with the potential to eliminate diagnostic testicular biopsies and improve the confidence of azoospermia diagnosis [10,124].

A comparative analysis of protein expression profiles of SP in normozoospermic, asthenozoospermic, oligozoospermic and azoospermic men has also been carried out by using two-dimensional differential gel electrophoresis (2-D DIGE) [101]. In this study, eight proteins—FN1, ACPP, proteasome subunit α type-3 (PSA-3), β-2-microglobulin (B2MG), LGALS3BP, PIP and cytosolic nonspecific dipeptidase (CNDP2)—showed a significantly higher expression in azoospermic patients, compared to the other groups. Most of these proteins (FN1, ACPP, B2MG, LGALS3BP and PIP) are localized in the extracellular region and are mostly associated with protein binding. Other studies have also suggested a role for PIP as a fertility biomarker, with variations in the levels of different isoforms depending on the fertility state of the studied men [134]. These data suggest a potential for these proteins as biomarkers of azoospermia.

In a further study, Rolland et al. [100] found that TKTL1, PGK2 and LDHC were specifically expressed in testicular germ cells. TKTL1 was expressed throughout germ cell development, with spermatogonia exhibiting the strongest labeling, whereas LDHC was expressed in both meiotic spermatocytes and post-meiotic spermatids and PGK2 was specifically expressed in elongated spermatids. These proteins were clearly detected in SP from fertile donors but were either undetectable or barely detectable in SP from vasectomized men and from patients with NOA or OA, suggesting a potential role as biomarkers of the status of the seminiferous epithelium and a signal of the stage of germ cell maturation arrest in infertile patients.

All in all, these findings demonstrate that seminal proteins are very useful and constitute a promising field to find a biomarker able to differentiate NOA from OA as well as to determine the origin of the male infertility disorder.

#### 3.3.2. SP Candidate Biomarkers for Asthenozoospermia, Oligozoospermia and Teratozoospermia

The general characteristics of normal and subfertile semen have been established by the World Health Organization (WHO, 2010) [135] for a systematic analysis of semen quality. Approximately 40–60% of infertile males present some deficiencies in at least one of the seminal parameters, particularly in sperm concentration, motility and/or morphology. Asthenozoospermia is considered when <32% of the SPZ have progressive motility; oligozoospermia is characterized by a sperm concentration or total number of ejaculate sperm below 15 or 39 millions of sperm cells respectively, and teratozoospermia is diagnosed when <4% of the sperm have normal morphology (WHO, 2010) [135]. Although any of these deficiencies may occur individually, the alteration of the three seminal parameters appears frequently combined in a single individual, e.g., oligozoospermia is often accompanied by poor motility (asthenozoospermia) and abnormal morphology (teratozoospermia) being then named oligo-astheno-terato-zoospermia.

Proteomic analysis of SP from asthenozoospermic patients provides a rich source of biomarker candidates for male infertility, proposing that functional abnormalities of the epididymis and prostate can contribute to asthenozoospermia. SP of healthy donors and asthenozoospermic patients was analyzed by using SDS-PAGE-LC–MS/MS, identifying 741 proteins [129]. Of these, 45 proteins were upregulated and 56 proteins were downregulated in asthenozoospermia compared to normozoospermic men. The proteomic changes in the seminal fluid from asthenozoospermic patients appear mainly associated with metabolism and energy production. Interestingly, altered sperm proteins in asthenozoospermia were also associated with disturbances in the generation of precursor metabolites and energy, being the citrate cycle the most affected pathway [6]. The most overexpressed proteins were enzymes, particularly those involved in proteolytic processes, while the most deregulated proteins were related to the regulation of reactive oxygen species (ROS). Among the identified proteins, the protein/nucleic acid deglycase DJ-1, a protein that reduces oxidative stress, was highly downregulated in asthenozoospermic patients and was proposed as a candidate biomarker for this condition. The decreased expression of DJ-1 was accompanied by an increase in ROS levels, a factor associated with male infertility [136].

Wu et al. [89] analyzed the human SP proteome from three normozoospermic and three asthenozoospermic individuals by using tandem mass tag (TMT) for peptide labeling and LC–MS/MS and described 29 differentially expressed proteins among the 524 proteins identified. Compared to normozoospermic samples, 22 proteins were downregulated including LDCH, SORD, ANXA2 and Kallikrein-2 (KLK2), and 7 proteins were upregulated in the asthenozoospermic group, including the heat shock-related 70 kDa protein 2 (HSP72). An altered expression of proteins involved in protein folding and degradation has been observed in previous proteome analyses in asthenozoospermic men and most of these proteins are heat shock proteins (HSPs), molecular chaperones that mediate protein folding and prevent protein aggregation. The exact relationship between altered expression of HSP and impaired motility is not fully understood, but several studies have linked HSP to male infertility [137]. In the study by Wu et al., [89] most differentially expressed proteins in asthenozoospermia were located in the extracellular space and the most significantly enriched molecular functions were enzyme-related functions, such as peptidase regulator activity and peptidase activity. The most enriched biological processes were proteolysis, secretion, and oxidation–reduction. LDCH, an enzyme that appears to have a key role in sperm capacitation, showed the highest decrease in asthenozoospermia and was also greatly under-expressed in NOA [100], suggesting that it could be a good candidate biomarker for male infertility.

Other proteins, dysregulated in patients with teratozoospermia, oligozoospermia and oligoteratozoospermia have been described in comparison with normozoospermic men [117]. Among them, Mucin-6 (MUC6), Orosomucoid 1 precursor (ORM1) and acid epididymal glycoprotein-like isoform 1 precursor, a CRISP1 homolog, were downregulated in teratozoospermic patients; tissue inhibitor of metalloproteinase (TIMP) and AZGP1, a protein involved in regulation of sperm motility, were upregulated while CLU and LGALS3BP were downregulated in patients with oligozoospermia; KLK3 and SEMG1 were upregulated in patients with oligoteratozoospermia; and LTF was upregulated in teratozoospermic patients, either with normal or low sperm count. These results show that various seminal proteins with important functions in SP are altered in patients with alterations in sperm number and/or morphology. The protein DJ-1 is absent in oligoteratozoospermic patients and differentially expressed in the other groups, confirming the validity of this protein as an infertility biomarker in these patients [129].

In the case of oligoasthenozoospermia, a study by Giacomini et al. [123] reported an underexpression of NPC2 and LGALS3BP and an overexpression of lipocalin-1 (LCN-1) and a PIP form in SP of oligoasthenozoospermic patients, in comparison with normozoospermic subjects. Liu et al. [115] analyzed the SP from 10 oligoasthenozoospermic patients and 10 normozoospermic patients using LC-MALDI-TOF-MS. They identified 734 proteins and found a series of differentially expressed proteins in oligoasthenozoospermic patients (22 upregulated proteins and 20 downregulated proteins). Upregulated proteins included LTF, glutathione S-transferase P (GSTP1), N-acetylglucosamine-1-phosphotransferase subunit gamma (GNPTG), tetraspanin-1 (TSPAN1), PIP, cathepsin L1 (CTSL), glycodelin (PAEP) and TGM4. Downregulated proteins included NPC2, PTGDS, KLK3, lysosome-associated membrane glycoprotein 2 (LAMP2), fibromodulin (FMOD), ECM1, AZGP1, CRISP1, epididymal sperm-binding protein 1 (ELSPBP1), extracellular superoxide dismutase [Cu-Zn] (SOD3) and ATP synthase subunit alpha (ATP5A1). The altered proteins in oligoasthenozoospermia mainly reflect a disturbance in development, metabolism, transport, antioxidation and immune response. For some of these proteins, i.e., NPC2, PIP, PTGDS, LTF, AZGP1, ECM1 and KL3, the results confirm the observations of previous studies [103,123,124,127,128], suggesting a promising utility as biomarkers of male fertility status.

Seminal proteome has also been studied in semen samples with deficiencies in sperm mitochondrial activity, acrosome integrity and DNA fragmentation by using LC–MS/MS [138]. In comparison with control samples, the altered proteins were mainly related to oxidoreductase activity, aminoglycan catabolism, endopeptidase inhibition, lysosomes and acute-phase response. The higher expression of protease inhibitors may conduce to a lower protection of sperm cells against proteolysis while the increased aminoglycan catabolism might impair sperm capacitation, as glycosaminoglycans are essential for sperm capacitation. Three potential biomarkers of sperm mitochondrial activity alterations were proposed: annexin A7 (ANXA7), glutathione S-transferase Mu3 (GSTM3) and endoplasmic reticulum resident protein 44 (ERP44), which were overexpressed in these patients. Interestingly, these biomarkers are related to the acrosome reaction, mitochondrial integrity and oxidative stress protection, demonstrating that the SP proteome correlates with the specific functional sperm alteration analyzed.

In samples with acrosome damage, the main enriched functions have been related to phospholipase inhibition, arachidonic acid metabolism, exocytosis, regulation of acute inflammation, response to hydrogen peroxide and lysosomal transport [138]. The enrichment of arachidonic acid metabolism and phospholipase inhibition might be associated with induction of a premature acrosome reaction and/or with sperm acrosome abnormalities. An overexpressed protein, phospholipid transfer protein (PLTP), was proposed as biomarker of SPZ presenting acrosome damage [138].

Enriched functions in samples with DNA fragmentation were related to prostaglandin biosynthesis and fatty acid binding with an underexpression of proteins related to extracellular matrix assembly and disassembly, metabolism of carbohydrates, metallopeptidase activity and negative regulation of peptidase activity. These results suggest a correlation between prostaglandin biosynthesis and sperm DNA fragmentation, which might be closely related to semen oxidative stress. On this basis, cysteine-rich secretory protein LCCL domain-containing 1 (CRISPLD1) and cysteine-rich secretory protein LCCL domain-containing 2 (CRISPLD2) were proposed as biomarkers of low DNA fragmentation, and retinoic acid receptor responder protein 2 (RARR2) and proteasome subunit alpha type-5 (PSMB5) as biomarkers of high DNA fragmentation [138].

The observation of a correlation between the SP proteome and the specific functional sperm alteration analyzed is a very interesting finding that reflects the high potential of SP as a source of biomarkers that may help to identify the causes of male infertility [35,138].

While many proteomic studies have been performed on whole SP, Garcia-Rodriguez et al. [22] analyzed the proteome of isolated human seminal exosomes and characterized the differences between protein expression in normozoospermic and non-normozoospermic subjects. A total of 1282 proteins were identified and, among them, 45 proteins showed a differential expression between the two groups. Most proteins underexpressed in dyspermic samples were related to energy production pathways, including LDCH, hexokinase-1 (HK1), purine nucleoside phosphorylase (PNP), adenine phosphoribosyltransferase (APRT) and solute carrier family 2 facilitated glucose transporter member 3 (SLC2A14). Other important proteins downregulated in non-normozoospermic samples were ELSPBP1 and CRISP1. In contrast, histone H2B type 1-A (HIST1H2B), a protein with an essential role in sperm DNA organization, was overexpressed. On the other hand, many proteins increased in the normozoospermic exosomes, such as beta-microseminoprotein (MSMB), myeloperoxidase (MPO), MIF and KLK2, are related to motility and cell adhesion. Previous studies have also shown an underexpression of CRISP1, LDCH, ELSPBP1 and MIF in infertile patients [34,111,113,116,118,124,139]. The important role of seminal exosomes in modulating sperm motility and capacitation in men with severe asthenozoospermia has been recently verified [34]. In this study, seminal exosomes were isolated from SP of normo-, astheno- and azoospermic patients and in vitro incubated with sperm cells. Incubation with exosomes derived from normozoospermic men, but not from asthenozoospermic individuals, improves SPZ motility, triggers capacitation and stimulates the acrosome reaction, probably by a transfer of different factors from exosomes to SPZ, including CRISP1, that was underexpressed in exosomes from asthenozoospermic men. On the contrary, incubation with exosomes from patients with severe asthenozoospermia causes a significant reduction of the percentage of progressive SPZ, with a very limited effect on capacitation and without effects on the acrosome reaction.

The proteomic profile of seminal exosomes from normozoospermic men and severe asthenozoospermic patients has also been recently evaluated by nano-scale liquid chromatographic tandem mass spectrometry with electrospray ionization (nLC–ESI–MS/MS) [23]. The analysis identified a total of 2138 proteins, with 89 proteins differentially expressed between exosomes of both groups. Among them, CRISP1, sperm-associated antigen 11B (SPAG11B) and defensin B126 (DEFB126), which are known to positively regulate sperm-specific functions, were among the proteins most strongly enriched in exosome samples from normozoospermic men; conversely, PAEP, a protein with an important role in the control of sperm capacitation, epididymal secretory protein E3-alpha (EDDM3A), likely involved in sperm maturation, and TGM4, that catalyzes the cross-linking of proteins and the conjugation of polyamines to specific proteins in sperm, were highly overexpressed in exosomes from asthenozoospermic samples.

#### 3.3.3. SP Candidate Biomarkers for Semen with High Reactive Oxygen Species (ROS) Levels

Oxidative stress has been implicated in the aetiology of male infertility [136]. It is known that human SPZ generate reactive oxygen species (ROS) in physiological amounts, which play a role in sperm functions during sperm capacitation, acrosome reaction and oocyte fusion [140]. However, an excessive production of ROS results in seminal oxidative stress with male infertility as a consequence [85,136]. As previously described, SP contains many proteins and metabolites that coat and protect SPZ from oxidative stress. In fact, downregulation of proteins involved in oxidative stress, such as DJ-1, is observed in the SP proteome of patients with asthenozoospermia and increased levels of ROS [117,129].

Proteomic studies conducted by Sharma et al. [111] identified 14 differentially expressed proteins in SP of patients with high (ROS+) or normal (ROS-) levels of ROS. Three proteins were uniquely expressed in the ROS- group: FN1 isoform three preprotein/FN1 isoform b precursor, MIF and LGALS3BP while a group of four preproteins were uniquely expressed in the ROS+ group: cystatin S (CST4) precursor, ALB preprotein, LTF precursor-1 peptide and PSA isoform 4 preprotein. All these ROS+ specific proteins were present in their precursor form, probably indicating some failure in post-translational mechanisms. Other proteins upregulated in the ROS+ group include PIP, SEMG2 precursor and ACPP short isoform precursor. Overexpression of PIP in human SPZ has been correlated with poor sperm quality and has also been described in azoospermic patients [101,103]. Analysis of the molecular functions revealed that the proteins uniquely expressed in the ROS- groups were mainly involved in antioxidant and proteolytic activities, functions that were diminished in the ROS+ group, probably leading to an increase in sperm apoptosis and death. In this context, a panel of 24 overexpressed proteins was identified in patients with idiopathic oliagoasthenoteratozoospermia and high oxidative stress. These proteins—mainly related to metabolism and inflammation, defense and stress responses—include: α-1-antichymotrypsin (AACT), a protein involved in the specific inactivation of serine proteases and control of oxidative stress; aldose reductase (ALDR), a NADPH-dependent oxidoreductase; diacylglycerol kinase eta (DGK), also involved in the regulation of oxidative stress and tubulin-folding cofactor B (TBCB), a protein involved in the assembly of α/β-tubulin heterodimers that is also increased in other tissues in response to oxidative stress. A further study by Intasqui et al. [141] also describes a high number of seminal proteins dysregulated in semen with high lipid peroxidation levels and proposed mucin-5B (MUC5B) as a potential biomarker related to oxidative stress.

The results of these studies clearly confirm the important role of ROS and semen oxidative stress on male infertility [91,136,141], suggesting a strong correlation with the seminal proteome analyzed in both normal and infertile patients. In this regard, infertile patients with different ROS levels (low, normal and high) showed divergent seminal proteome in their ejaculates, in comparison with fertile individuals, with the identification of 44 unique proteins in seminal ejaculates with high ROS levels, relative to controls, and a gradual decrease in extracellular proteins across the ROS gradient [87]. In general, proteins involved in metabolic processes, energy production, protein folding and degradation were overexpressed and those involved in acute inflammatory responses were underexpressed in the three groups of infertile patients with increased ROS levels. Interestingly, the cytokine family with sequence similarity 3 (FAM3D), also known as metabolism regulating signaling molecule D, disappeared in all infertile patients with increased ROS levels and was only expressed in fertile donors. Additionally, there was an increase in various proteases in all groups of infertile men, particularly ADAMTS-1 and MME. The protease ADAM metallopeptidase with thrombospondin type 1 motif 1 (ADAMTS-1) is a heparin-binding protein that belongs to the A disintegrin and metalloprotease (ADAM) family that also includes fertilin (ADAM2), one of the first proteins implicated in integrin-mediated sperm–egg binding [142]. Studies in mice have shown that ADAMTS1-null animals display several developmental abnormalities, primarily within the urogenital systems, affecting normal growth, organ morphology and function and female fertility [87].

A pronounced increase of the membrane metallo-endopeptidase MME was also found in infertile patients with increased levels of ROS. MME, also known as neprilysin, neutral endopeptidase (NEP), cluster of differentiation 10 (CD10) and common acute lymphoblastic leukemia antigen (CALLA), is a zinc-dependent metalloprotease that cleaves peptides at the amino side of hydrophobic residues and inactivates several peptide hormones including enkephalins, tachykinins and bradykinins [109,110]. MME appears to be essential for the development and reproduction in mammals and has a high activity in seminal fractions, in comparison to other tissues [143]. Subiran et al. [109] and Pinto et al. [110] have reported the presence of MME in the prostasomes and in the neck region of SPZ, suggesting a role in the regulation of sperm function.

#### 3.3.4. SP Candidate Biomarkers for Varicocele

Varicocele, the abnormal tortuosity and dilatation of the veins of the pampiniform plexus within the spermatic cord, is one of the leading causes of male infertility and has a detrimental effect on spermatogenesis by inducing a state of testicular hyperthermia, hypoxia and oxidative stress [144]. Infertile men with varicocele have poor semen quality, seminal oxidative stress and sperm DNA damage [145]. The finding of SP biomarkers associated with varicocele might be of great interest to provide an early indication for varicocelectomy before the appearance of clinical symptoms.

Several studies have shown that different seminal proteins associated with normal physiological function of SPZ are differentially expressed in varicocele patients. Fariello et al. [113] analyzed protein expression in smoking patients with varicocele using 2D-SDS-PAGE and LC–MS/MS and identified 20 differentially expressed proteins. Among them, four proteins were found exclusively in the control group: the serine protease inhibitor SERPINA1 (also known as A1AT), the acid ceramidase ASAH1, the clathrin-assembly lymphoid myeloid leukemia protein CALM and the epididymal glycoprotein CRISP1. These proteins are involved in the inflammatory response, regulation of proteolysis, sperm maturation and sperm–oocyte fusion and may explain some of the male infertility problems associated with varicocele. A study in non-smoking varicocele patients should clarify the influence of smoking on these results.

Zylbersztejn et al. [120] used 2-DE followed by electrospray mass spectrometry (ESI-QuadTOF-MS) to analyze the human SP of adolescents with and without varicocele. They found that adolescents with varicocele and normal semen quality (VNS) show an overexpression of spermatogenesis proteins, whereas adolescents with varicocele and abnormal semen quality (VAS) show an overexpression of apoptosis-regulated proteins, compared to adolescents without varicocele. Specifically, there was an overexpression of SEMG1 and an underexpression of KLK3 and ACPP in the SP of varicocele patients. NPC2 and PIP were only detected in the control group whereas the serine/threonine protein kinase SMG1 and insulin-like growth factor binding protein-3 (IGFBP-3) were exclusively found in adolescents with clinical varicocele, suggesting a potential use as biomarkers of impaired spermatogenesis in this disease. Additionally, in adolescents with varicocele, the same group reported an increase in SP levels of insulin-like growth factor-binding protein 7 (IGFBP7), a protein related to cell differentiation and proliferation, and a decrease in deoxyribonuclease-1 (DNase I), responsible for apoptosis regulation, particularly in patients with varicocele and altered semen analysis [121]. The increase in the proliferative and inflammatory activity accompanied by a dysregulation of apoptosis may thus contribute to the sperm functional alterations and DNA fragmentation observed in these patients [120,121]. Del Giudice et al. [119] compared the seminal proteome of 23 healthy adolescents and 54 adolescents with varicocele, 37 of them with normal semen parameters and 17 with altered semen parameters. A total of 541 proteins were identified. Among them, a series of proteins were proposed as biomarkers: 45 kDa calcium-binding protein (CAB45), left–right determination factor 1 (protein lefty-1), DNase I and lipid phosphate phosphohydrolase 1 (PLPP1) associated with controls; IBP-7 and Ig γ-3 chain C region (HDC) associated with the VNS group and cysteine-rich secretory protein 3 (CRISP3) increased in the VAS group.

Proteomic studies by Panner Selvam et al. [112,126,144] have also found an overexpression of proteins related to oxidative stress, particularly peroxiredoxin-1 (PRDX1) and peroxiredoxin-2 (PRDX2), in the SP of varicocele patients, compared to fertile controls. Proteins associated with the sperm–oocyte interaction, such as T-complex protein 1 subunit alpha isoform a (TCP1), T-complex protein 1 subunit delta isoform a (CCT4) and T-complex protein 1 subunit theta isoform 1 (CCT8); protein folding, such as HSPA2 and lipid peroxidation and DNA fragmentation, such as apolipoprotein A2 (APOA2), were absent or underexpressed in varicocele patients [126]. Other proteins associated with normal sperm functions, such as SEMG1 and SEMG2, acrosin (ACR), elongation factor 1-gamma (EEF1G) and PGK2 were downregulated in SP from varicocele, probably explaining the poor sperm motility and concentration usually found in these patients [126].

In summary, different proteomic analyses show that SP of controls is enriched in proteins related to antioxidant activity and homeostatic maintenance of sperm cell function, while SP from varicocele patients is impoverished in proteins that protect from apoptosis and is enriched with proteins related to the inflammation and immune response, leading to a chronic inflammatory reaction and a decreased semen quality in these subjects [119,120,121,126,144].

Differences in the seminal proteome profile of patients with unilateral and bilateral varicocele have also been described [112]. Panner Selvam et al. [146] observed a dysregulation of exosome-associated proteins in patients with unilateral varicocele, compared to fertile controls, with an overexpression of ANXA2 and an underexpression of kinesin-1 heavy chain (KIF5B). Compared to unilateral varicocele patients, inflammatory response pathways were dysregulated in bilateral varicocele, with an underexpression of alpha-1-acid glycoprotein 1 (ORM1), alpha-1-acid glycoprotein 2 (ORM2), SERPINA1, GGT1, apolipoprotein D (APOD) and FN1, and an overexpression of ECM1, polymeric immunoglobulin receptor (PIGR) and ALDR. The underexpressed proteins in bilateral varicocele were mostly included in the inflammation interleukin 6 (IL-6) signaling and inflammation Janus kinase-signal transducer and activator of transcription (Jak-STAT) pathways. Proteins associated with oxidative stress (PRDX2), DNA fragmentation (fatty acid synthase; FASN) and the inflammatory response (FN1) were also overexpressed in the bilateral varicocele group, indicating an increased inflammation and oxidative stress in this group, compared to unilateral varicocele. On this basis, these last three proteins were proposed as potential non-invasive SP biomarkers for the differentiation of unilateral and bilateral varicocele patients.

#### 3.3.5. SP Candidate Biomarkers for Primary and Secondary Infertility

A proteomic profiling of sperm in men with primary and secondary infertility was carried out by Intasqui et al. [125] suggesting large proline-rich protein (BAG6) and HIST1H2BA as potential biomarkers of male infertility. In addition, a recent report has shown the existence of differences between the seminal proteomes of patients with primary infertility (absence of previous pregnancies) and secondary infertility (at least one previous successful pregnancy) [114]. The authors performed a large-scale proteomic analysis using 1D-PAGE and LC–MS/MS and identified 48 and 53 differentially expressed proteins in the primary and secondary infertility groups, respectively, compared to the control fertile group. Of particular interest, there was an overexpression of ANXA2 and CDC42 and an underexpression of SEMG2 in men with primary infertility and an overexpression of ANXA2 and amyloid precursor protein (APP) in secondary infertility. The overexpression of ANXA2 in both primary and secondary infertility, and the results of earlier studies, showing an altered expression of this protein in SP and prostasomes of subfertile or infertile men [89,146,147] suggest its potential utility as a biomarker for both primary and secondary infertility. On the other hand, CDC42 and SEMG2 can be useful candidate biomarkers for primary infertility and APP for secondary infertility. A dysregulated expression of semenogelins has also been reported in SP of men with abnormal semen parameters [111,117] and varicocele [120,126,146], confirming the importance of these proteins as markers of infertility. Importantly, this study shows that seminal proteins might permit the identification of the main cause associated with primary and secondary infertility [114].

## 4. Conclusions and Future Perspectives

The importance of SP has been neglected for years, considering it as a mere vehicle for sperm transport and protection. On the contrary, SP is particularly enriched in proteins, RNAs and lipids, encapsulated in exosomes—vesicles able to convey to SPZ an impressive repertoire of new secretory products.

Recent large-scale proteomic studies have characterized the proteomic landscape of SP that is very complex and has a great potential as a source of candidate biomarkers for diagnosis and differentiation of male fertility disorders. Interestingly, SP proteins, particularly those associated with exosomes, have been involved in the control of important sperm functions; furthermore, their differential expression in the case of azoospermia, asthenozoospermia, oligozoospermia and teratozoospermia or other conditions of male infertility has allowed the identification of a wide variety of biomarker candidates, prompting to deepen the biological pathways in which they are involved. However, the results of these studies also indicate that the SP proteomic component needs further analysis, mostly focused on determining the organ of origin of the different proteins, as it should reflect the physiological/pathological state of the testis, the epididymis and/or the different accessory sex glands. Additionally, the differences between studies suggest the need to standardize collection procedures for SP, as well as the method of sample preparation.

Currently, research aims to suggest that SP is able to modulate the maternal environment at conception and that both sperm- and SP-specific mechanisms may participate in how a father influences the embryo development as well as the long-term health of his offspring, with important transgenerational consequences. In the era of ‘omics studies’, the proteomic analysis of SP may surely help to expand the picture of potential biomarkers suggested as useful diagnostic tools in fertility assessment and in assisted reproductive technology (ART) success prediction. This still remains an intriguing new point for future research. Similarly, studies concerning the impact of a paternal lifestyle, air pollution and age on genetic, epigenetic and proteomic signature of sperm is, currently, growing up [148]; however, the role played by SP in such a direction is largely overlooked.

A more comprehensive understanding of the above-mentioned mechanisms may surely help in a better diagnosis of male infertility as well as the management of seminal exosomes—useful natural nanocarriers—to target drug delivery may be a novel intriguing therapeutic approach.

## Figures and Tables

**Figure 1 ijms-21-07022-f001:**
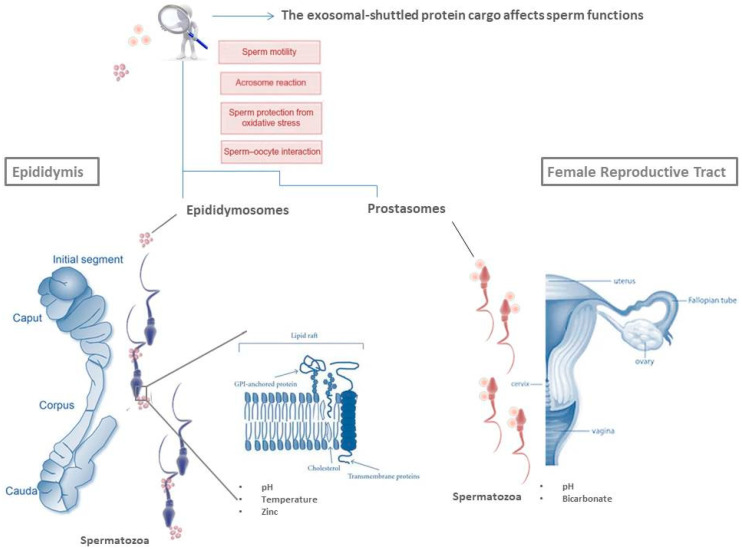
The fate of spermatozoa (SPZ) beyond testis. During their transit along the epididymis, SPZ receive proteins carried by epididymosomes. The GPI-anchor-mediated docking is a way of epididymosomal delivery cargo from lipid rafts of epididymosomes to lipid rafts of SPZ. Other conditions, such as pH, temperature and zinc concentration surely influence vesicle fusion. During ejaculation, SPZ encounter another subset of exosomes: the prostasomes. The recruitment of prostasomes by SPZ requires the presence of bicarbonate and an environment with a favorable pH. Proteins shuttled via exosomes are involved in important sperm functions such as sperm motility, prevention of premature acrosome reaction, protection of SPZ from oxidative stress and sperm–oocyte interaction.

**Table 1 ijms-21-07022-t001:** Representative SP proteins proposed as biomarkers of male infertility.

Protein	Infertility Relationship	References
Albumin preprotein (ALB precursor)	Exclusively expressed in patients with high levels of ROS	[111]
Aldose reductase (ALDR)	Increased in patients with oliagoasthenoteratozoospermia and high levels of ROS	[111]
	Highly increased in bilateral varicocele	[112]
Alpha-1-acid glycoprotein 1 (ORM1),	Highly decreased in bilateral varicocele	[112]
Alpha-1-acid glycoprotein 2 (ORM2)	Highly decreased in bilateral varicocele	[112]
α-1-antichymotrypsin (AACT)	Increased in patients with oliagoasthenoteratozoospermia and high levels of ROS	[111]
α -1-antitrypsin (SERPINA1)	Highly decreased in bilateral varicocele	[112]
	Absent in varicocele	[113]
Apolipoprotein D (APOD)	Highly decreased in bilateral varicocele	[112]
Annexin 2 (ANXA2)	Increased in primary infertility	[114]
	Decreased in asthenozoospermia	[89]
ATP synthase subunit alpha (ATP5A1).	Decreased in oligoasthenozoospermia	[115]
Cathepsin L1 (CTSL)	Increased in oligoasthenozoospermia	[115]
CDC42	Increased in primary infertility	[114]
Clusterin (CLU)	Highly decreased in azoospermia	[116]
	Decreased in oligozoospermia	[117]
Cysteine-rich secretory protein 1 (CRISP1)	Highly decreased or absent in OA	[116,118]
	Decreased in oligoasthenozoospermia	[115]
	Decreased in dyspermia	[22]
	Decreased in asthenozoospermia	[23,34]
	Absent in varicocele	[113]
Cysteine-rich secretory protein 3 (CRISP3)	Increased in varicocele	[119]
Deoxyribonuclease-1 (DNase I)	Decreased in varicocele	[119,120,121]
Dipeptidase 3 (DPEP3)	Expressed exclusively in fertile controls	[102,108]
Diacylglycerol kinase eta (DGK)	Increased in patients with oliagoasthenoteratozoospermia and high levels of ROS	[111]
Epididymal secretory protein E1 (NPC2)	Highly decreased or absent in azoospermia	[116]
	Absent in OA	[122]
	Decreased in oligoasthenozoospermia	[115,123]
	Absent in varicocele	[120]
Epididymal secretory protein E3-alpha (EDDM3A)	Highly increased in asthenozoospermia	[23]
Epididymal sperm-binding protein 1 (ELSPBP1)	Decreased in oligoasthenozoospermia	[115]
	Decreased in dyspermia	[22]
Extracellular matrix protein 1 (ECM1)	Decreased in oligoasthenozoospermia	[115]
	Highly increased in bilateral varicocele	[112]
ECM1 and Testis-expressed sequence 101 (TEX101)	Diagnosis and differentiation of OA and NOA. Differentiation of NOA subtypes	[10,124]
Family with sequence similarity 3 (FAM3D)	Absent in patients with high levels of ROS	[87]
Fatty acid synthase (FASN)	Highly increased in bilateral varicocele	[112]
Fibromodulin (FMOD)	Decreased in oligoasthenozoospermia	[115]
Fibronectin (FN1)	Increased in azoospermia	[103]
	Highly increased in bilateral varicocele	[112]
Fibronectin I isoform 3 preprotein/fibronectin 1 isoform b precursor (FN1 precursor)	Absent in patients with high levels of ROS	[111]
Galectin-3 binding protein (LGALS3BP)	Decreased in oligozoospermia	[117]
	Decreased in oligoasthenozoospermia	[123]
	Increased in azoospermia	[103]
	Absent in patients with high levels of ROS	[111]
Glutathione hydrolase 1 proenzyme (GGT1)	Highly decreased in bilateral varicocele	[112]
Gammaglutamyltransferase 7 (GGT7)	Expressed exclusively in NOA	[108]
Glutathione S-transferase P (GSTP1)	Increased in oligoasthenozoospermia	[115]
Glycodelin (PAEP)	Highly increased in oligoasthenozoospermia	[115]
	Highly increased in asthenozoospermia	[23]
Heat shock-related 70 kDa protein 2 (HSP72)	Increased in asthenozoospermia	[89]
Histone H2B type 1-A (HIST1H2BA)	Expressed exclusively in fertile controls	[102,108,125]
	Increased in dyspermia	[22]
Insulin-like growth factor binding protein-3 (IGFBP-3)	Expressed exclusively in clinical varicocele	[120]
Insulin-like growth factor-binding protein 7 (IGFBP7)	Increased in varicocele	[119,121]
Kallikrein-2 (KLK2)	Decreased in dyspermia	[22]
	Decreased in asthenozoospermia	[89]
L-lactate dehydrogenase C chain (LDHC)	Expressed exclusively in fertile controls	[100,102,108]
	Highly decreased in asthenozoospermia	[89]
	Decreased in dyspermia	[22]
Lactotransferrin (LTF)	Increased in teratozoospermia	[117]
	Increased in oligoasthenozoospermia	[115]
Lactotransferrin precursor-1 peptide (LTF precursor)	Exclusively expressed in patients with high levels of ROS	[111]
Lipocalin-1 (LCN-1)	Increased in oligoasthenozoospermia	[123]
Macrophage migration inhibitory factor (MIF)	Decreased in dyspermia	[22]
	Absent in patients with high levels of ROS	[111]
Membrane metallo-endopeptidase (MME)	Highly increased in patients with high levels of ROS	[87]
N-acetylglucosamine-1-phosphotransferase subunit gamma (GNPTG)	Increased in oligoasthenozoospermia	[115]
Orosomucoid 1 precursor (ORM1)	Decreased in teratozoospermia	[117]
Peroxiredoxin-1 (PRDX1)	Increased in varicocele	[126]
Peroxiredoxin-2 (PRDX2)	Increased in varicocele	[126]
	Highly increased in bilateral varicocele	[112]
Phosphoglycerate kinase 2 (PGK2)	Expressed exclusively in fertile controls	[100,102,108]
	Decreased in varicocele	[126]
Polymeric immunoglobulin receptor (PIGR)	Highly increased in bilateral varicocele	[112]
Prolactin-inducible protein (PIP)	Increased in oligoasthenozoospermia	[115]
	Increased in oligoasthenozoospermia	[123]
	Decreased in NOA	[122]
	Increased in azoospermia	[103]
	Increased in patients with high levels of ROS	[111]
	Absent in varicocele	[120]
Prostaglandin-H2-D-isomerase (PTGDS)	Decreased in oligoasthenozoospermia	[115]
	Highly decreased in OA	[108,127,128]
Prostate-specific antigen (KLK3)	Increased in oligoteratozoospermia	[117]
	Decreased in oligoasthenozoospermia	[115]
	Decreased in varicocele	[120]
Prostate-specific antigen isoform 4 preprotein (KLK3 precursor)	Exclusively expressed in patients with high levels of ROS	[111]
Prostatic-specific acid phosphatase (ACPP)	Increased in azoospermia	[103]
	Decreased in varicocele	[120]
Prostatic-specific acid phosphatase short isoform precursor (ACPP precursor)	Increased in patients with high levels of ROS	[111]
Prostate-specific transglutaminase 4 (TGM4)	Increased in oligoasthenozoospermia	[115]
	Highly increased in asthenozoospermia	[23]
Protein/nucleic acid deglycase (DJ-1)	Absent in oligoteratozoospermia	[117]
	Highly decreased in asthenozoospermia	[129]
Semenogelin-1 (SEMG1)	Increased in oligoteratozoospermia	[117]
	Decreased in varicocele	[126]
Semenogelin-2 (SEMG2)	Decreased in varicocele	[126]
	Decreased in primary infertility	[114]
Semenogelin-2 precursor (SEMG2 precursor)	Increased in patients with high levels of ROS	[111]
Serine/threonine protein kinase (SMG1)	Expressed exclusively in clinical varicocele	[120]
Serum amyloid p-component (SAP)	Decreased in azoospermia, absent in patients with Sertoli cell-only syndrome	[116]
Sorbitol dehydrogenase (SORD)	Expressed exclusively in NOA	[108]
	Decreased in asthenozoospermia	[89]
Extracellular Superoxide dismutase [Cu-Zn] (SOD3)	Decreased in oligoasthenozoospermia	[115]
Superoxide-dismutase (SOD)	Decreased in azoospermia	[116]
Tetraspanin-1 (TSPAN1),	Increased in oligoasthenozoospermia	[115]
Transketolase-like protein 1 (TKTL1)	Expressed exclusively in fertile samples	[100]
Tubulin-folding cofactor B (TBCB)	Increased in patients with oliagoasthenoteratozoospermia and high levels of ROS	[111]
Zinc alpha-2 glycoprotein 1 (AZGP1)	Increased in oligozoospermia	[117]
	Decreased in oligoasthenozoospermia	[115]

Abbreviations: NOA non-obstructive azoospermia; OA obstructive azoospermia; ROS reactive oxygen species.

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
