# Peer review of "Exosome Composition and Seminal Plasma Proteome: A Promising Source of Biomarkers of Male Infertility"

_ijms, 2020, doi:10.3390/ijms21197022_

Round 1
Reviewer 1 Report
The paper by Candenas and Chianese summarises literature on seminal plasm proteome and its potential to offer novel biomarkers for altered spermatogenesis and male infertility.
The review is complete and useful and presents also a nice introduction on exosomes contained in seminal plasma. My suggestion is to avoid the mere lists of proteins and make an effort to present, when possible, a mechanism linking the lack or the presence of certain proteins and the tested infertility condition. The paper gives much importance to possible use of seminal plasma proteins as biomarkers for male infertility. If the Authors like, such concept could be included in the title.
Specific comments:
Lines 115-117. The end of the sentence is nor clear, please rephrase it
Lines 163-167 and line 173. Check English
Line 192-3. The meaning of this technical sentence can be not understandable by every reader. Please rephrase it.
Line 222. I suggest factors instead of tools
Figure 1 could be enriched by other details, for instance the possible mechanisms where the proteins carried by exosomes are involved
Paragraph 3.1 could be substituted by a table, as it is a simple list of proteins
Line 320. What the Authors mean with cellular proteins, all proteins are “cellular”
Lines 321-3. Already written and can be deleted.
Lines 328-9. This finding could be expanded in view of the important role of proteases/proteases inhibitors
LINE 363. Which are “the two pools”?
Lines 494-6. It is not clear to which patients, the sentence is referred.
Line 583, please indicate the meaning of the acronym
Line746 and following. The review does not report on patients treated by ARTs and on predictive power of their outcome. Similarly, nothing has reported on the influence of seminal plasma on conception, embryo development and health of offspring. Please rephrase this part by clearly presenting the concepts as future perspectives.
Author Response
The review is complete and useful and presents also a nice introduction on exosomes contained in seminal plasma. My suggestion is to avoid the mere lists of proteins and make an effort to present, when possible, a mechanism linking the lack or the presence of certain proteins and the tested infertility condition. The paper gives much importance to possible use of seminal plasma proteins as biomarkers for male infertility. If the Authors like, such concept could be included in the title.
We are very grateful to the Reviewer for his comments. We have followed his suggestions as much as possible and changed the title of the review to include the concept of biomarkers. It has been really a hard task to organise this review, due to the thousands of proteins present in seminal plasma and the high number described as candidate biomarkers. In this context, the chapter 3 of the review has been divided considering the biomarkers described for each particular infertility disorder and, within each disorder, the physiological mechanisms that are overexpressed or underexpressed, which derives from the differentially expressed proteins. The paragraph 3.1 may be a mere list of proteins, and can be removed, if the reviewer considers it the best, but it indicates the origin of the most abundant proteins present in seminal plasma and this origin will be essential, in the future, for diagnosis and differentiation of male infertility disorders.
However, considering that the centrepiece of the review is the description of the proteomic landscape of the human seminal plasma and its potential role as indicator of sperm dysfunctions, this concept has been included in the title.
Specific comments:
Lines 115-117. The end of the sentence is nor clear, please rephrase it
The sentence has been rephrased.
Lines 163-167 and line 173. Check English
English has been checked and sentences have been rephrased for greater clarity
Line 192-3. The meaning of this technical sentence can be not understandable by every reader. Please rephrase it.
This sentence has been omitted in the revised version of the manuscript in accordance to reviewer 2’s suggestions.
Line 222. I suggest factors instead of tools
“Tools” has been changed in “Factors” as suggested
Figure 1 could be enriched by other details, for instance the possible mechanisms where the proteins carried by exosomes are involved
Figure 1 has been enriched adding the main sperm functions that are influenced by the exosomal-shuttled protein cargo, as suggested by the reviewer
Paragraph 3.1 could be substituted by a table, as it is a simple list of proteins
We have tried to substitute the paragraph 3.1 by a table, as suggested by the Reviewer, with the result of a 2-pages table. However, the inclusion of these proteins in a table increases the length of the manuscript and does not permit to include the details and the differential abundance of the described proteins. As indicated above, although it is a list of proteins, it indicates the origin of the different proteins, which is a very important factor in the identification of biomarker candidates and in the differentiation of male infertility disorders (see for example Drabovich et al., Nat Rev Urol 2014). For these reasons, the paragraph 3.1 has been maintained in the revised version of the manuscript.
Line 320. What the Authors mean with cellular proteins, all proteins are “cellular”
The bioinformatic analysis used in large-scale proteomic studies assign a probable cellular component and the cellular annotation indicates a probable association with exosomes. Nevertheless, we agree completely with the Reviewer and the text has been corrected and the term “cellular” omitted (see page 8, lines 372 in the revised version of the manuscript).
Lines 321-3. Already written and can be deleted.
The sentence in these lines referred to the most abundant proteins identified in the particular study of Pilch and Mann (Gen Biol 2006). The sentence has been deleted in the revised version of the manuscript
Lines 328-9. This finding could be expanded in view of the important role of proteases/proteases inhibitors
We agree with the Reviewer´s consideration and the particular role of proteases/protease inhibitors in seminal plasma could be the subject of a next review. We are unable to include this in the present review due to its general character and the observation that it is already too extensive.
LINE 363. Which are “the two pools”?
The text has been modified to indicate which are the two pools in the revised version of the manuscript
Lines 494-6. It is not clear to which patients, the sentence is referred.
The text has been corrected to clarify that it refers to asthenozoospermic patients
Line 583, please indicate the meaning of the acronym
The meaning of the acronym has been included in the text (see page 9 lines 392 in the revised version of the manuscript)
Line746 and following. The review does not report on patients treated by ARTs and on predictive power of their outcome. Similarly, nothing has reported on the influence of seminal plasma on conception, embryo development and health of offspring. Please rephrase this part by clearly presenting the concepts as future perspectives.
Thanks a lot to reviewer for his comments. Actually, no data concerning the influence of seminal plasma on conception, embryo development and health of offspring are discussed in this review.
Our aim was just to throw food for thought, as the reviewer has perfectly understood.
With this in mind, we completely rephrase conclusion, presenting the concepts as future perspectives
We are very grateful to the Reviewer for his constructive comments, that have been very useful to improve the manuscript.
Reviewer 2 Report
The review contains 2 main chapters:
1) Exosomes in the male reproductive tract and 2) Proteomic landscape of human seminal plasma in relationship to the male infertility
In my opinion, the review is too extensive. I recommend the authors to shorten the Introduction part, which should be more focused on humans.
Reviews by other authors are too much cited in the Introduction and Exosomes sections, authors should use more primary sources.
I would also recommend shortening the part of Exosomes. A description of the epididymis is not necessary. The authors should really focus in this section only on proteins in exosomes and target humans. This section contains interesting information, but the authors should keep only those that are relevant to the topic of the human SP proteome.
Figure 1 should be better placed, not at the end of the chapter and with a better description directly in the text with reference to the figure.
The extensive Chapter 3 is conveniently structured and elaborated. Introduction to the chapter 3.3. should contain summary information about Table 1 directly in the text. The authors should consider about the title of this chapter (3.3.), in my opinion it is not entirely apposite. Likewise, the subtitle of table “Infertility condition”, rather “relationship”.
In addition, the title of the review is not entirely appropriate. It should be said that the review concerns biomarkers of infertility as well as exosomes.
The Abstract should focus more on the mentioned reproductive dysfunctions discussed in chapter 3.3. The same should be mentioned in the Conclusion.
Minor comments:
l.24 spermatozoa have tail or flagellum - please select one term
l.31 Littre's glands
l.165-166 please provide another reference, e.g. Plant 2016
Please, add protein abbreviations to the text, they are missing somewhere, leave them in parentheses outside the reference, e.g. l.186 (PSCA) [69]
Author Response
The review contains 2 main chapters:
1) Exosomes in the male reproductive tract and 2) Proteomic landscape of human seminal plasma in relationship to the male infertility
In my opinion, the review is too extensive. I recommend the authors to shorten the Introduction part, which should be more focused on humans.
Thanks a lot to reviewer for his comments. We have considerably shortened the introduction part as suggested, eliminating misleading details.
Reviews by other authors are too much cited in the Introduction and Exosomes sections, authors should use more primary sources.
As suggested, reviews by other authors have been largely changed in primary sources, in both Introduction and Exosomes sections.
I would also recommend shortening the part of Exosomes. A description of the epididymis is not necessary. The authors should really focus in this section only on proteins in exosomes and target humans. This section contains interesting information, but the authors should keep only those that are relevant to the topic of the human SP proteome.
As suggested, the part of exosomes has been shortened. An extremely brief description of the epididymis - concerning its three different segments - still remains for the purpose of emphasizing the segmental production of a heterogeneous population of epididymosomes, with a different lipid and protein composition. This aspect is far from a deep description of the epididymis that, we agree, is not necessary for the purpose of the paragraph.
Other details concerning the presence of non-coding RNAs or chromosomal DNA in SP vesicles have been omitted as suggested, because not relevant to the topic of the human SP proteome.
Figure 1 should be better placed, not at the end of the chapter and with a better description directly in the text with reference to the figure.
As suggested, figure 1 has been placed inside the paragraph 2, not at the end, with a more convenient description along the text
The extensive Chapter 3 is conveniently structured and elaborated. Introduction to the chapter 3.3. should contain summary information about Table 1 directly in the text. The authors should consider about the title of this chapter (3.3.), in my opinion it is not entirely apposite. Likewise, the subtitle of table “Infertility condition”, rather “relationship”.
We are very grateful to the Reviewer for his comments. We agree completely with the Reviewer and have changed the title of chapter 3.3 which is now: SP candidate biomarkers of male infertility. The same has been made with subsequent subchapters of the 3.3 chapter. We have also included information about Table 1 in the text of introduction to chapter 3.3.
The subtitle of Table 1 has also been modified following the suggestion by the Reviewer.
In addition, the title of the review is not entirely appropriate. It should be said that the review concerns biomarkers of infertility as well as exosomes.
We have changed the title of the review including both concerns “biomarkers of infertility” and “exosomes”
The Abstract should focus more on the mentioned reproductive dysfunctions discussed in chapter 3.3. The same should be mentioned in the Conclusion.
Abstract has been changed mentioning reproductive dysfunctions discussed in chapter 3.3. Conclusion has been completely rephrase, also in accordance to reviewer 1’s suggestions.
Minor comments:
l.24 spermatozoa have tail or flagellum - please select one term
We select “tail” term
l.31 Littre's glands
Changed accordingly
l.165-166 please provide another reference, e.g. Plant 2016
The suggested reference has not been found. For this reason, it has not been changed as required.
Please, add protein abbreviations to the text, they are missing somewhere, leave them in parentheses outside the reference, e.g. l.186 (PSCA) [69]
The text has been carefully revised to add protein names and or protein abbreviations, when possible. Sometimes, it is very difficult to assign a symbol, due to the existence of many different members of a same protein family and the lack of identification of the particular member involved, at least in the literature we have reviewed. In some of these cases, we have decided to omit the names of the proteins in the revised version of the manuscript, in other cases, we have modified the text to indicate that we make reference to a family, for example, in the case of globulins. In other cases, for example, in the case of inhibin-B, there is no abbreviation, as it is a complex of alpha (INHA) and beta b (INHBB) subunits. The PSCA abbreviation has been located outside the reference, as indicated by the Reviewer.
We are really very grateful to the Reviewer for his constructive comments, that have been very useful to improve the manuscript.
Round 2
Reviewer 2 Report
The authors have satisfactorily modified the text according to my comments.
I think the manuscript is now suitable for publishing in IJMS.